# Nuclear Magnetic Resonance Relaxation Pathways in Electrolytes for Energy Storage

**DOI:** 10.3390/ijms241210373

**Published:** 2023-06-20

**Authors:** Carla C. Fraenza, Steve G. Greenbaum, Sophia N. Suarez

**Affiliations:** 1Physics Department, Hunter College, City University of New York, 695 Park Avenue, New York, NY 10065, USA; cf1884@hunter.cuny.edu (C.C.F.); sgreenba@hunter.cuny.edu (S.G.G.); 2Physics Department, The Graduate Center, City University of New York, 365 Fifth Avenue, New York, NY 10016, USA; 3Physics Department, Brooklyn College, City University of New York, 2900 Bedford Avenue, Brooklyn, NY 11210, USA

**Keywords:** NMR spin-lattice relaxation time, NMR fast field cycling (FFC), electrolytes

## Abstract

Nuclear Magnetic Resonance (NMR) spin relaxation times have been an instrumental tool in deciphering the local environment of ionic species, the various interactions they engender and the effect of these interactions on their dynamics in conducting media. Of particular importance has been their application in studying the wide range of electrolytes for energy storage, on which this review is based. Here we highlight some of the research carried out on electrolytes in recent years using NMR relaxometry techniques. Specifically, we highlight studies on liquid electrolytes, such as ionic liquids and organic solvents; on semi-solid-state electrolytes, such as ionogels and polymer gels; and on solid electrolytes such as glasses, glass ceramics and polymers. Although this review focuses on a small selection of materials, we believe they demonstrate the breadth of application and the invaluable nature of NMR relaxometry.

## 1. Introduction

The need for clean and sustainable energy storage and conversion devices has increased exponentially in the last few decades with the advancement of technology and with concerns about dwindling natural resources and environmental degradations [1,2,3]. One of the main drivers of this increase has been the worldwide focus on common communications consumer products such as mobile phones, tablets and laptops, which are mainly dependent on energy storage devices such as batteries. Another driver of this increase has been renewable energy sources such as wind and solar, which are the fastest growing electricity producers [4,5,6] with an average increase of 6.3% annually. Although renewable energy sources are weather-dependent, the main problem with their increased implementation has been the dissimilarity between their peak production and actual usage-by-consumers times. Because of this, energy storage devices are needed to maximize the efficiency and reliability of renewable energy sources.

Energy storage and conversion devices have differences in their functionalities as well as their ions or species transports. Devices such as batteries and supercapacitors function to produce and store electrical energy through the conversion of chemical energy, while fuel cells and solar cells function to convert chemical and light energy into immediately deliverable electrical energy [2,3,7]. Within each category of devices there are further delineations. For example, the solid oxide fuel cell (SOFC) operates at high temperatures (600–1000 °C) and relies on the transport of gaseous species through porous ceramics, while the polymer electrolyte fuel cell (PEFC) operates at lower temperatures (<100 °C) and relies on species transport through multiple dissimilar layers and materials.

In terms of storage, the most effective devices are secondary batteries, of which the most prevalent have been lithium-ion batteries (LIBs) [4,5,6,8,9,10,11]. Made available by Sony in 1991, they initially powered smaller mobile devices because of their high energy density and longer lifetime compared to other storage devices. However, with the vast number of improvements made since, they have become ubiquitous the world over, powering larger consumer needs such as electric vehicles. Despite this, serious problems still plague LIBs, mainly safety and cost. Another electrochemical energy storage device worth mentioning is the supercapacitor. Whereas batteries have the advantage of high energy densities, supercapacitors [1,2,3,12,13,14,15,16,17]—although lacking in this regard—have higher power densities and longer life cycles. They also have wide temperature ranges and require lower maintenance compared to LIBs [14], which oftentimes require battery management systems (BMS). They came into commercial application in 1957 when General Electric patented the first electric double layer capacitor (EDLC), in which charge storage is accomplished by the electrostatic separation of charges at the interfaces of the electrodes and electrolyte. Other supercapacitors include the pseudocapacitor—in which charge storage is accomplished through Faradaic processes—and the hybrid capacitor, which is a combination of the EDLC and pseudocapacitor and accomplishes charge storage through both Faradiac and non-Faradaic processes. As mentioned above, supercapacitors suffer from low energy densities, and improving this requires the right combination of electrodes and electrolyte.

This review focuses specifically on the use of Nuclear Magnetic Resonance (NMR) relaxometry to probe dynamics in electrolyte energy conversion and storage devices. The importance of the review is highlighted in the details provided about relaxometry, the various processes of relaxation and the information revealed by relaxometry in the various studies depicted. Although NMR has been heavily used to study dynamics, most of these studies have focused on self-diffusion coefficients, not relaxometry. Because of this, relaxometry has been less used despite its many attributes. This review aims to showcase relaxometry and the wealth of details that can be obtained from its application.

## 2. Electrolytes

At the center of these electrochemical storage and conversion devices is an electrolyte, which acts simultaneously as an ion conductor and electron insulator between the electrodes. Its electron-insulating property ensures the charge transfer processes responsible for producing electricity take place at the electrode–electrolyte interfaces, while its ion-conducting property is directly related to the devices’ efficiency [18,19]. The electrolyte can also affect the device’s cycle stability, electrochemical window of operation, capacity, safety, and other operating conditions [18,19]. In LIB electrolytes, one of the main functionalities of the electrolyte is the formation of the solid electrolyte interphase (SEI) layer [20,21,22,23,24,25,26,27,28] at the anode. In general, the SEI layer passivates the electrodes and contributes to the Li-ion diffusion processes necessary for energy production. The formation of the SEI layer is a dynamic process that depends on several factors including the type, components and concentrations of the additives and the bulk electrolyte and the specific anode material, its structure and associated surface chemistry. It involves the transformation from initially formed inorganic (LiF and Li_2_CO_3_) [22,23] and organic (ex. Lithium ethylene decarbonates (LEDC)) [20,21] species, and its compositional state evolves through numerous ageing processes [24,25,26,27], such as hydrolysis, reactions between Li salt and intermediate decomposition products, thermal decomposition and continuous electrolyte reduction [29]. Because of the multifaceted dependence of the composition and performance of the SEI layer, the combination of electrodes (especially the anode) and electrolyte must be a merger based upon electrochemical, thermal and mechanical compatibilities.

There are various types of electrolytes, including liquids, solids or quasi-solid state, and redox active. Each type of electrolyte is further delineated into subcategories. For example, within the liquid electrolytes are aqueous and non-aqueous types, while the redox electrolytes include gels, ionic liquids, organic and aqueous types.

Organic liquid electrolytes have been the most applied in secondary battery applications [20,21,24,25,26,27,30,31,32,33,34,35,36,37]. The basic component of organic electrolytes are organic solvents, and some of their properties should include high dielectric constants (>15) which facilitate salt dissociation and reduce ion pairing. The organic solvent should also have low viscosity, which enhances high ionic mobility. Additionally, the solvent should be electrochemically stable in its interactions with the electrodes. In the case of LIBs, liquid electrolytes are usually a combination of a lithium salt such as lithium hexafluorophosphate (LiPF_6_), lithium tetrafluoroborate (LiBF_4_), lithium perchlorate (LiClO_4_) or lithium trifluoromethanesulfonate (LiCF_3_SO_3_), dispersed in one or more non-aqueous solvents such as dimethyl carbonate (DMC), diethyl carbonate (DEC), propylene carbonate (PC) or ethylene carbonate (EC). These organic carbonates are generally flammable and volatile, and when combined with alkali salts such as LiPF_6_, which have limited thermal stability and are sensitive towards hydrolysis, the result can be unsafe for consumers due to heat generation, thermal runaway, cell venting, fire and even explosion. The inclusion of film-forming additives such as fluoroethylene carbonate (FEC) and vinylene carbonate (VC) and flame-retardant additives such as trimethyl phosphate (TMP) and triethyl phosphate (TEP) offer improvements in the electrolytes by providing more stable SEI layers and lowered flammability of the organic solvents [38,39,40].

Ionic liquids (ILs) have also gained increasing attention as electrolytes and electrolyte solvents in the last decade [41,42,43,44,45,46,47,48,49,50]. ILs possess low volatility, low to zero flammability, wide electrochemical ranges (up to 6 V) and fairly good ionic conductivities. The use of pure ILs as electrolytes has been hindered by their relatively high solution viscosities. The incorporation of organic additives to ILs has been shown to improve the mixtures’ ionic conductivities, and the use of additives can offer more effective SEI layers [44]. The most used ILs consist of imidazolium-based cations [43,44,45,46,47,48,49,50] such as imidazolium, pyridinium, piperidinium and pyrrolidinium, all of which are classified as aprotic. The most commonly used anions include bis(trifluoromethylsulfonyl)amide (CF_3_SO_2_)_2_N^−^), hexafluorophosphate (PF_6_) and tetrafluoroborate (BF_4_). On the other hand, deep eutectic solvents (DESs) [36,37,51,52,53,54,55,56], which sometimes are considered a sub-class of ILs due to their similar properties, have shown to be promising as electrolytes as well [36,37,51,52,53,54,55,56]. These solvents are defined as a mixture of two or more components, which may be solid or liquid, and that at a particular composition present a high melting point depression, becoming liquids at room temperature. Unlike typical ILs, DESs can be inexpensive, readily synthesized and can often be prepared from biodegradable and nontoxic constituents.

Semi-solid-state electrolytes (SSEs) have attracted interest because they combine the main advantages of liquid and solid electrolytes, namely the high ionic conductivity and mechanical stability, respectively. Among them are solid polymer electrolytes (SPEs), ionogels (IGs) and gel polymer electrolytes (GPEs). IGs consist of an IL electrolyte confined in a mesoporous inorganic matrix which macroscopically behaves as a solid, but at the nanoscale a liquid-like state is maintained [57,58,59,60,61,62], while GPEs consist of a gelled polymer matrix swollen in a liquid electrolyte [63,64]. Due to their optimal interfacing with electrodes and liquid-like transport behavior, GPEs have found application in commercial LIBs [65,66]. Although less investigated compared to liquid electrolytes, SPEs offer several advantages over liquids due to their flexibility, adhesion to electrodes, volumetric compactness, general accommodation of volumetric changes in the electrodes and safety [67,68,69,70]. This is why solid-state batteries are generally not affected by ionic short circuits and can therefore be assembled into higher voltage battery stacks without external separation between each cell [71]. The most studied type of SPE is based on poly (ethylene oxide) (PEO), into which a salt has been dissolved [72,73,74,75,76]. However, its electrolytes generally suffer from very low ionic conductivities (≤10^−6^ S/cm) and low transport numbers (t_Li+_~0.2–0.3) [77,78] due to the strong coordination of lithium ions with the oxygen atoms in the PEO backbone [79,80,81]. In spite of this, studies on PEO persist [82,83,84,85], with researchers focused on modifications that will enhance Li^+^ transport.

## 3. NMR Relaxation

NMR relaxation refers to the process by which thermodynamic equilibrium is restored to its radiofrequency-exited magnetization [86,87,88,89,90,91,92]. Relaxation studies began in the 1940s and are one of the most important investigative tools in the study of dynamics in materials. This is because overall NMR is non-destructive, nucleus-specific and can elucidate dynamics spanning frequencies from the Hz to MHz range. Despite these factors, NMR suffers from a low signal-to-noise (S/N) ratio and a correspondingly low resolution due to a combination of factors. Firstly, NMR is a radiofrequency-dependent technique, which means it is limited to smaller sizes of magnetizations. Secondly, magnetization (or signal) intensity also depends on nucleus-specific factors such as natural abundance, which further limits the magnetization size and corresponding Larmor frequency. Finally, sample sizes are generally limited to at most 1 mL, further compounding the S/N ratio. For a more expansive discussion on NMR the interested reader is referred to the various cited references [86,87,88,89,90,91,92].

The relaxation times most determined are spin-lattice (T_1_), spin-spin (T_2_) and the Nuclear Overhauser Effect (NOE) [86,87,88,89,90,91,92]. Regardless of the type, spin relaxation is caused by temporal fluctuating interactions and the associated relaxation time depends on the strength and type of interactions, as well as the spectral density functions (J(ω)). These temporal fluctuations are from the local magnetic field, or in the case of quadrupole nuclei—the local electric field gradient (EFG)—at the site of the affected spin. Fundamentally, the spectral density functions provide information on the power available for causing spin transitions among the different frequencies [90]. More expressly, it is the Fourier Transform of the time-dependent correlation function G(τ), which is associated with a correlation decay time constant (τ_c_) for the associated dynamical process (see Equation (1)).
(1)$$J\left(\omega \right)=\;A\int_{-\infty }^{\infty }G\left(\tau_{c}\right)e^{-{i\omega \tau }_{c}}{d\tau }_{c}$$

Here A represents a normalization coefficient, and ω represents the angular frequency in rads/s. Simplistically, τ_c_ is defined as the time in which a molecule remains in any given position before a collision causes it to change position or orientation. Generally, rapid fluctuations result in small τ_c_, while slow fluctuations have larger τ_c_. Since the focus of this paper is on the applications of relaxation times, we will provide a brief review of those applications most commonly studied. However, for a more in-depth treatise on spin relaxation times and the associated mechanisms, the interested reader is directed to the various references cited [86,87,88,89,90,91,92].

Spin-lattice (T_1_), or longitudinal relaxation, is the process by which the radiofrequency-excited net spins or the magnetization component along the quantization field (M_z_) returns to thermal equilibrium. In T_1_ relaxation, the spins give up their non-equilibrium energy obtained from the radiofrequency pulses to the molecular system or lattice, thereby returning the spin system to the required unequal Boltzmann distributions. Because M_z_ is determined by the quantized nuclear spins, T_1_ relaxation is determined only by transitions that change the nuclear spin state and energy [87]. T_1_ is generally determined through the saturation or—more often used—inversion recovery pulse sequences. In the inversion recovery pulse sequence (π-τ-π/2), two radiofrequency pulses (π, and π/2) are separated by varying delay times (τ). Immediately following the π pulse, the magnetization is inverted. During the time interval, the magnetization relaxes back to thermal equilibrium at a rate of T_1_^−1^. The application of the π/2 causes the magnetization to rotate from the z-axis into the x-y plane, where the signal may be picked up by a receiver. This process is depicted more clearly in Figure 1.

For an increasing array of τ values, the signal intensity as a function of τ may be fitted and the T_1_ extracted. The recovery of M_z_ to equilibrium is governed by the equation:(2)Mz(τ)=|Mo|(1−2e−τT1),where M_o_ and τ represent the equilibrium magnetization, as per the Curie law, and the time elapsed after the completion of the radiofrequency excitation field. The saturation recovery pulse sequence consists of a train “comb” of high-power π/2 pulses separated from the detection π/2 pulse by a delay τ. The purpose of the pulse train is to saturate the magnetization or equalize the populations of the spin energy states. The delay time τ must be greater than or on the order of the spin-spin relaxation time of the signal to allow complete acquisition of the signal. The detection pulse then brings the magnetization into the x-y plane for detection. The saturation recovery method has some advantages over inversion recovery, but the most important is the reduction in measurement time, thereby making it more suitable for spin systems with long T_1_ (i.e., T_1_ > 10 s). For an increasing array of τ values, the signal intensity as a function of τ may be fitted to Equation (3) and the T_1_ extracted.
(3)$$M_{z}\left(\tau \right)={\;M}_{0}\left[1-e^{\frac{-\tau }{T_{1}}}\right]$$

If the fluctuations in the local magnetic field or the EFG are due to thermally activated motions, then the temperature- or magnetic field- (ergo resonance frequency ω_0_ in rad/s) dependent T_1_ can provide information on the motional process. Associated with this process is a rate constant R_1_ = 1/T_1_ that provides details on motions with rates similar to the Larmor frequency, and when measured as a function of temperature allows the determination of the activation energies of dynamical processes. Moreover, when R_1_ is measured as a function of frequency, it provides straightforward access to the frequency-dependent spectral density J(ω) according to [93]:(4)$$R_{1}\left(\omega \right)=K\left[J\left(\omega \right)+4J\left(2\omega \right)\right],$$
where K is a constant that describes the strength of the dominating spin interaction, which can be dipolar (K = A_DD_) or quadrupolar (K = A_Q_).

Spin-spin (also termed transverse or T_2_) relaxation is the process of transverse decoherence. A simple visualization of this process is the decay of the magnetization in the transverse plane of a perfectly homogeneous quantization field following a ${\left(\pi /2\right)}_{x}$ radiofrequency excitation pulse. Since the transverse components of the equilibrium magnetization (M_x_ and M_y_) are the result of spins precessing with nonrandom phases, transverse relaxation can be affected by any interaction that results in phase changes, including isoenergetic ones [87]. T_2_ is generally determined through an echo pulse sequence, the type of which depends on conditions such as homogeneity of the static magnetic field and the length of T_2_. Commonly used pulse sequences include the Hahn echo (π/2-τ-π-τ-acquire), the Carr Purcell Meiboom Gill (CPMG, π/2-τ-[π-τ-acquire-τ]_n,_ n represents iterations) and the Stimulated echo (π/2-τ_1_-π/2-τ_2_–(π/2)_φ_-τ_1_–acquire). Regardless of the echo pulse sequence used, the attenuation profile is as depicted in Figure 2.

Of the three, the CPMG provides the most direct determination of T_2_, for which the decoherence is governed by the equation:(5)Mx,y(t)=|Mo| e−tT2

Associated with this process is a rate constant R_2_ = 1/T_2_ which—in the case of a perfectly homogeneous quantization field and Lorentzian spectrum—is related to the line width at half maximum $\upsilon_{1/2}$ [86] through the relationship $R_{2}={\;\pi \upsilon }_{1/2}$, where $\upsilon_{1/2}$ is the width of NMR spectrum (Fourier transform of the NMR signal) at half of its maximum amplitude. R_2_ provides details on motions at zero frequency.

The Nuclear Overhauser Effect (NOE) refers to the net change in the signal intensity of the enhanced spin I due to the relaxation of the perturbed spin S in a dipole-dipole (DD) coupled spin system [88]. Since the NOE depends on the DD interaction, modulating fluctuations due to molecular motion can produce cross relaxation and magnetization transfer. Like the DD interaction, the NOE depends on the inverse sixth power of the internuclear distance ($r_{\mathrm{IS}}^{-6}$) between spins I and S. Because of this, neighboring spins must be within 5 Å for NOE applications. Associated with the NOE are the cross-relaxation rate (σ_IS_) and dipolar longitudinal relaxation (ρ_IS_) rates, given by:(6)$$\sigma_{\mathrm{IS}}={\left(\frac{\mu_{o}}{4\pi }\right)}^{2}\frac{\hbar^{2}\gamma_{I}^{2}\gamma_{S}^{2}}{10}\;\left(\frac{6\tau_{c}}{1+{\left(\omega_{I}+{\;\omega }_{S}\right)}^{2}\tau_{c}^{2}}-\frac{\tau_{c}}{1+{\left(\omega_{I}-{\;\omega }_{S}\right)}^{2}\tau_{c}^{2}}\right)r_{\mathrm{IS}}^{-6},$$
(7)$$\rho_{\mathrm{IS}}={\left(\frac{\mu_{o}}{4\pi }\right)}^{2}\frac{\hbar^{2}\gamma_{I}^{2}\gamma_{S}^{2}}{10}\;\left(\frac{6\tau_{c}}{1+{\left(\omega_{I}+{\;\omega }_{S}\right)}^{2}\tau_{c}^{2}}+\frac{3\tau_{c}}{1+{\;\omega }_{I}^{2}\tau_{c}^{2}}+\frac{\tau_{c}}{1+{\left(\omega_{I}-{\;\omega }_{S}\right)}^{2}\tau_{c}^{2}}\right)r_{\mathrm{IS}}^{-6}$$

Here γ_I_ and γ_S_ are the magnetogyric ratios, ω_I_ and ω_S_ are the Larmor frequencies of spins I and S, respectively, and $\mu_{o}$ is the magnetic permeability of vacuum. Although initially intended for intramolecular spins, the NOE has recently been applied to intermolecular spin interactions in highly viscous ILs media [94,95,96,97]. This is in spite of additional modulations of the intermolecular DD interaction by the translational motions of the molecules. Examples of these will be discussed.

The methods discussed so far rely on a single fixed static magnetic field B_o_ (or frequency). However, it is also possible to measure frequency- (ergo field) dependent T_1_ (1/R_1_) through the fast field cycling (FFC) relaxometry technique [98,99]. By its nature, FFC is a low-field, low-resolution technique that covers a broad range of magnetic fields from ^1^H Larmor frequencies of a few kHz to about 10 MHz in a single experiment. As previously stated, T_1_ is affected by molecular motions. Therefore, being able to access information over a broad frequency range can provide invaluable details about molecular dynamics.

At its foundation, the technique requires a variable magnetic field B_o_ that is capable of fast switching times and maintaining good static homogeneity at the desired fields. A schematic of the technique is shown in Figure 3. The sample is initially polarized at a higher magnetic field (B_p_), resulting in a magnetization with its spins distributed in the associated energy levels according to the Boltzmann equilibrium distribution. After some time, the field is lowered to the relaxation field (B_r_), causing the magnetization to re-populate according to the new Boltzmann equilibrium distribution as required amongst the various energy levels. This decreased magnetization resulting from its relaxation in the lower field is then measured using a π/2 radiofrequency (RF) pulse or echo sequence during the detection magnetic field (B_d_). It must be stated that the detection field setting B_d_ is constant during the experiment, and it is in fact the field that determines the Larmor frequency of the nucleus.

The decay in magnetization is characterized by a spin-lattice relaxation time (T_1_), and by using an array of increasing B_r_ values, the full spin-lattice relaxation dispersion curve or spectral density can be obtained. If the equilibrium magnetization at the polarization (B_p_) and relaxation (B_r_) fields are represented as M_o_(B_p_) and M_o_(B_r_), respectively, then the magnetization M_z_(τ) detected after the relaxation interval time τ is given by Equation (8):(8)$$M_{z}\left(\tau \right)={\;M}_{o\;}\left(B_{r\;}\right)+\left[M_{o\;}\left(B_{p\;}\right)-{\;M}_{o\;}\left(B_{r\;}\right)\right]e^{\frac{-\tau }{T_{1}}}.\;$$

The accuracy of the measurements is dependent on several key instrument factors. These include fast field switching times, high sensitivity, and good field homogeneity. High sensitivity requires that both the polarization and detection fields be as high as possible with good thermal stability. Currently, magnetic field strengths of 3 T corresponding to ^1^H Larmor frequency of 125 MHz are available. Switching times should be no more than a few milliseconds to ensure the achievement of the acceptable field inhomogeneity of <<10^−4^ over a sample volume of 1ml. A fast switching time is also desired because it places a lower limit on the T_1_ that can be detected. For more details on the specifics of the instrumentation requirements and the technique, the interested reader is encouraged to explore the references cited [98,99].

A major limitation of the FFC technique is the signal-to-noise (S/N) ratio. Due to the hardware requirements of fast switching times with good homogeneity and thermal stability, signal averaging as is the norm in high-resolution NMR is very limited. The single transient S/N for high resolution NMR is directly related to B_o_^3/2^ where B_o_ is the static field. For field cycling NMR, however, S/N is dependent directly on both the polarization and detection fields, adding to the requirement of having both as high as possible to achieve good sensitivity.

## 4. Translational and Rotational Dynamics in Liquid and Confined Electrolytes

In studies of the dynamics in liquid ILs and LIB electrolytes, measurements of T_1_ have been the most ubiquitous. As previously stated, associated with the dynamic processes is a correlation function G(τ) (or distribution) and τ_c_ [86,87,88,89,90,91,92]. To properly analyze the dynamics, researchers often need to assume a priori what the responsible relaxation mechanism or mechanisms are. In general, there are five main mechanisms for spin relaxation:(9)$$\frac{1}{T_{1}}=\frac{1}{T_{1}^{\mathrm{DD}}}+\frac{1}{T_{1}^{\mathrm{CSA}}}+\frac{1}{T_{1}^{\mathrm{SR}}}+\frac{1}{T_{1}^{\mathrm{Sc}}}+\frac{1}{T_{1}^{Q}}\;,$$
where $T_{1}^{\mathrm{DD}}$, $T_{1}^{\mathrm{CSA}}$, $T_{1}^{\mathrm{SR}}$, $T_{1}^{\mathrm{SC}}$ and $T_{1}^{Q}$ represent the DD, chemical shift anisotropy (CSA), spin rotation (SR), scalar coupling (SC) and quadrupolar interactions (Q) relaxation times, respectively. In the case of liquids with spin I = ½ nuclei such as ^1^H and ^19^F, oftentimes intramolecular and intermolecular DD interactions are the dominant relaxation mechanisms. Both DD interactions are modulated in time by molecular rotations and the relative translational diffusion of molecules carrying the nuclear spins [91]. Because of this, the T_1_ profiles of ILs can be modeled by considering these two molecular motions. Spin rotation and scalar couplings become significant when small molecules are involved. In the case of quadrupolar nuclei such as ^2^H, ^7^Li, ^11^B and ^23^Na, the quadrupole interaction usually dominates.

The main model used in relaxation time analyses is the Bloemburgen, Purcell and Pound (BPP) [86,87,88,89,90,91,92,93]. In this model, the correlation function G(τ) is given the simple form of a time-dependent decaying exponential of the form ~exp(−t/τ_c_). When a plot of T_1_ versus inverse temperature is conducted, the BPP model predicts a symmetric v-shaped curve. However, when applied to highly viscous, quadrupolar nuclei or solid media, deviations arise due to the local distribution of interactions, thus resulting in the heterogeneous nature of the material. In these cases, the correlation function is better represented as a stretched exponential of the form ~exp(−t/τ_c_)^β^, where β represents the distribution of interaction strengths throughout the media and ranges between 0 and 1.

### 4.1. Ionic Liquids Dynamics

One of the main probes of ion dynamics in ILs has been through the use of dilute solute particles such as benzene, carbon monoxide (CO) and carbon dioxide (CO_2_). The rationale for this is that the solute (ergo probe) dynamics will be dependent upon its interactions with the cations and anions and that by using probes one can obtain a microscopic view of the ILs’ local structure. The following studies highlight this rationale.

Yasaka et al. [100] used ^17^O T_1_ to probe the rotational dynamics of CO in 1-ethyl-3-methylimidazolium bis(trifluoromethanesulfonyl)imide ([C_2_C_1_im][NTf_2_]), 1-butyl-3-methylimidazolium ([C_4_C_1_im] NTf_2_ and 1-methyl-3-octylimidazolium ([C_8_C_1_im]) NTf_2_. In the ‘extreme narrowing’ limit (ω_o_τ_R_ << 1) of the BPP model, the T_1_ for a quadrupolar nucleus exposed to an axially symmetric electric field gradient is given by:(10)$$\frac{1}{T_{1}}=\frac{3\pi^{2}}{10}\;\frac{\left(2I+3\right)}{I^{2}\;\left(2I-1\right)}\;{\left(\frac{e^{2}\mathrm{Qq}}{h}\right)}^{2}\tau_{R},$$
where $e^{2}\mathrm{Qq}$/h is the quadrupolar coupling constant (QCC). For all three ILs, $\tau_{R}$ was ≤ 10 ps and between 295–413 K it decreased by one order of magnitude due to the reduction in viscosity. The authors used the Stokes–Einstein–Debye (SED) equation,
(11)$$\tau_{R}=\frac{\eta \mathrm{fVC}}{k_{B}T},$$
to determine if there were deviations between the respective correlation times. Here V is the Van der Waals volume of the solute, C is the boundary condition parameter with values between C_slip_ (0) and C_stick_ (1), f is the shape parameter that accounts for the non-spherical nature of solutes, k_B_ is the Boltzmann constant, T is the absolute temperature and η is the solution viscosity. The results of the CO dynamics evaluated by the SED equation were 10 to 100 times faster than the predicted slip hydrodynamic values and were classified in the superslip regime. The importance of this result is that it separates the impediments to the rotational motion of the CO molecule from the ILs’ viscosity effects. The authors also investigated the effect of increasing alkyl chain length on the CO molecule’s rotational dynamics and determined that at fixed η/T values, $\tau_{R}$ decreased. In other words, as the size of the nonpolar domain increases, so does the CO molecule’s rotation. They also found a fractional SED relationship ($\tau_{R}\propto \;{\left(\eta /T\right)}^{p}$) between $\tau_{R}\;$and η/T, which the authors surmised was due to the Van der Waals volume difference between the solute and solvent molecules. Here the constant p represents the degree of deviation of $\tau_{R}$ from the SED relationship.

In a similar more expansive and recent study, Endo et al. [101] also used ^17^O T_1_ in their study of 10 NTf_2_^−^ anion-based ILs. The cations included four imidazolium- (C_1_C_0_im^+^, C_1_C_2_im^+^, C_1_C_4_im^+^, C_1_C_6_im^+^), four phosphonium- (P_2225_^+^, P_4448_^+^, P_44412_^+^, P_8884_^+^) and two ammonium- (N_4441_^+^, N_8884_^+^) based types. The T_1_ data all fell into the motional narrowing regime, and like the Yasaka study [100], the $\tau_{R}$ values were faster than the SED predictions with the slip boundary condition. Fits of $\tau_{R}$ vs. η/T were also non-linear, which the authors surmised was due to the free rotation-like motion of the CO molecules. The authors also found increasing deviation of $\tau_{R}$ from SED predictions with an increase in the number of carbon atoms in the alkyl chain, despite the cation type. CO molecules are expected to reside closer to the alkyl chains and results show that their rotation occurs within 1 ps, making them generally insensitive to the viscosity effects. This result supports the dissimilarities between local environments of the cations’ head groups and their alkyl chains.

In a related study, Rumble et al. [38] used ^2^H T_1_ to study the dynamics of dilute deuterated benzene (C_6_D_6_) in [C_4_C_1_im] tetrafluoroborate ([BF_4_]) and two deuterated versions of the C_2_C_1_im^+^ cation (C_2_C_1_im^+^-d_1_ and C_2_C_1_im^+^-d_6_) in C_2_C_1_im^+^ NTf_2_. The uniqueness of this study lies with the authors’ use of three different magnetic field strengths in which the ^1^H Larmor frequencies were 300, 400 and 850 MHz for the determination of the ^2^H T_1_ measurements. The purpose of this was to explore the applicability of the ‘extreme narrowing’ condition that is often used in analyzing ILs’ dynamics. Over the temperature range investigated (240–320 K), the authors found agreement between the extreme narrowing regimes at the three frequencies for all three probe molecules. Like the Yasaka et al. study [100], their calculated $\tau_{R}$ values for the C_6_D_6_ probe molecule expressed as a function of η/T also showed deviations from the hydrodynamic predictions and were instead related by the fractional SED. Through the combination of Molecular Dynamics Simulations (MDS) and the relaxation studies, the authors showed there were differences between the dynamics of the C_6_D_6_ probe molecule in conventional organic solvents and ILs. When spun about its six-fold axis, the dynamics was similar in both media and faster than predicted by the hydrodynamic predictions. However, the ‘tumbling’ rotation about in-plane axes was more impeded in ILs due to their higher viscosities, resulting in even greater deviations from the hydrodynamic predictions.

On the other hand, Strate et al. [102] studied the relationship between the rotational and translational dynamics of ions and hydrogen bond lifetimes to provide information on the structural properties of ILs. They focused on the IL (2-hydroxyethyl) trimethylammonium bis-(trifluoromethylsulfonyl)imide ([Ch][NTf_2_]) and measured the deuteron quadrupolar relaxation rates (^2^H R_1_) at 500 MHz (^1^H Larmor frequency) of the OD hydroxyl groups of the cation in the deuterated IL as a function of temperature as a way to calculate the reorientational correlation time $\tau_{\mathrm{OD}}\;$of this group. They used Equation (10) to obtain $\tau_{\mathrm{OD}}$, where the QCC was calculated by using its relationship with proton chemical shifts ^1^H δ; within the temperature range between 303 K and 406 K, the $\tau_{\mathrm{OD}}$ values decrease from 32.4 ps to 3.1 ps. This hydroxyl group is involved in the doubly ionic hydrogen bond with the oxygen of [NTf_2_]; therefore, the authors concluded that both modes, rotation and translation of the cation, were affected by the formation and lifetime of doubly ionic hydrogen bonds in the IL. Based on the good agreement between the measured and simulated reorientational correlation times as a function of temperature, they computed self-diffusion coefficients and hydrogen bond lifetimes by means of molecular dynamics simulation. Surprisingly, the magnitude of the self-diffusion coefficient of both cations and anions was found to be similar over the whole temperature range, suggesting that both species diffuse as hydrogen-bonded ion pairs. Additionally, the calculated hydrogen bond lifetime consisted of two separate time domains, namely, short- and long-time contributions. The short-time behavior was within the picosecond time range, showing time constants very similar to the computed reorientational correlation times. However, the long-time part was in the nanosecond range and was attributed to the exchange based on translational diffusion. Since the long-time contribution of the hydrogen bond lifetime amounts to about two-thirds of the overall lifetime, the authors concluded that long-living hydrogen-bonded ion pairs do exist in this ionic liquid.

The fast field cycling (FFC) relaxometry technique [98,99] has been widely used to study molecular motions of ions in ILs [47,62,103,104,105,106,107,108,109,110,111,112,113] due to its outstanding ability to reveal information about translational and rotational dynamics in only one experiment. Although FFC-NMR is a low-resolution technique, it is possible to study the dynamics of cations and anions independently if they are composed of different NMR active nuclei. For instance, if the IL composed of 1-butyl-3-methylimidazolium bis(trifluoromethylsulfonyl)imide ([BMIM][TFSI]) is considered, according to the chemical structure of the cation and anion, studying the relaxation rate of ^1^H and ^19^F is equivalent to studying the molecular dynamics of BMIM^+^ and TFSI^−^, respectively [62,104].

As previously stated, the T_1_ dispersions of ILs can be modeled by considering translational and rotational motions that modulate in time the intermolecular and intramolecular interactions. In general, intramolecular dipolar interactions fluctuate in time more rapidly compared to the intermolecular ones and would therefore be effective for relaxation at short time scales. This means its effect would show up in the high field (high Larmor frequency) regime of the spin-lattice relaxation profile. On the other hand, intermolecular interactions are mostly dependent on the translational motion of the whole molecule, which is slower, and therefore efficient for relaxation in the low field regime (low Larmor frequency). Then, in the limit of low Larmor frequencies, the relaxation rate can be expressed as $R_{1}\left(\nu \right)≅B-A\sqrt{\nu }$, where A and B are constants and ν = ω/2π. B represents the relaxation contribution from rotational dynamics and A is given by [114,115,116]:(12)A=(μ04π)2n γ4ћ2(82+215)(πD12)32.

Here µ_o_ is the permeability of free space, γ is the gyromagnetic ratio, n is the nuclear density and D_12_ is the relative diffusion coefficient defined as the sum of the self-diffusion coefficients of the involved molecular pair (D_12_ = D_1_ + D_2_). In the case of identical molecules, D_12_ is equal to two times the self-diffusion coefficient. Then, by plotting the spin-lattice relaxation rate values as a function of the square root of the Larmor frequency ν, R_1_(ν) vs. ν^1/2^ shapes into a linear function at lower frequencies and D_12_ can be easily calculated from the parameter A. The importance of this approximation is the fact that it does not depend on the model used to describe the complete relaxation profile and it allows the calculation of the diffusion coefficient of the ions in ILs, whose values have been shown to agree with the ones obtained with the standard pulsed field gradient (PFG-NMR) diffusion technique [47,103,111].

Although in general ILs consist of interacting non-spherical ions, the force-free-hard-sphere (FFHS) model [114,115] has been employed successfully to describe the spin-lattice relaxation due to translational diffusion in these systems [47,62,103,104,105,106,107,108,109,110,111,112,113,117,118]. Considering that there are no previous studies that extend this diffusion model for elongated molecules, it may be considered as a very first approximation to describe the translational diffusion in ionic liquids. This model is based on the assumptions that the molecular motion can be compared to that of a rigid sphere that obeys Fick’s diffusion equation, the spins are positioned in the center of the molecules and there is a uniform distribution of the molecules outside the distance of the closest approach d. This last assumption imposes a boundary value condition since spin–pair interactions are neglected for interspin distances that are shorter than the distance d. This excluded volume effect is treated by including an extra term in the relative diffusion equation, representing the potential from averaged forces between the spin-bearing molecules [114,115]. This model provides an approximate but improved description of the system as compared with the approach used by other authors [119], where the excluded volume effect was not considered. The general expression of $R_{1}$ for three-dimensional translational diffusion given by the FFHS model is as follows [114]:(13)$$R_{1\mathrm{Diff}}=\frac{108}{5}\left(\frac{\mu_{0}}{4\pi }\gamma^{2}h\right)\frac{1}{d^{3}}n\int_{0}^{\infty }\frac{u^{4}}{81+9u^{2}-2u^{4}+u^{6}}\left[\frac{\tau_{\mathrm{Diff}}}{u^{4}+{\left({\omega \tau }_{\mathrm{Diff}}\right)}^{2}}+\frac{4\tau_{\mathrm{Diff}}}{u^{4}+{\left(2{\omega \tau }_{\mathrm{Diff}}\right)}^{2}}\right]\mathrm{du}.$$

Here the correlation time $\tau_{\mathrm{Diff}}$ is defined as $\tau_{\mathrm{Diff}}=\frac{d^{2}}{D_{12}}$, where D_12_ is the relative diffusion coefficient and u is a dimensionless integration variable. This model can take a quite simple expression by assuming that the diffusion can be represented by finite jumps in the Brownian limit r^2^/6d^2^ → 0, where r^2^ is the mean square jump distance defined by r^2^ ≡ 6Dτ, τ is the mean time between jumps and D is the self-diffusion coefficient. Then, it is possible to arrive at the following expression for the homonuclear translational relaxation rate R_1_ [86,120]:(14)$$R_{1\mathrm{Diff}}^{\mathrm{ii}}\left(\omega_{i}\right)=\frac{A_{D}^{\mathrm{ii}}}{d_{\mathrm{ii}}D_{\mathrm{ii}}}\left[\tilde{J}\left(z(\omega_{i})\right)+4\tilde{J}\left(z(2\omega_{i})\right)\right],$$
where ω = 2πν and ν is the Larmor frequency; $A_{D}=\frac{8}{45}\pi \gamma^{4}{\text{ћ}}^{2}{\left(\frac{\mu_{0}}{4\pi }\right)}^{2}n$; $z\left(\omega \right)\equiv \sqrt{2{\omega d}^{2}/D_{12}}$, and $\tilde{J}(z)=\frac{1+\frac{5}{8}z+\frac{z^{2}}{8}}{1+z+\frac{z^{2}}{2}+\frac{z^{3}}{6}+\frac{z^{5}}{81}+\frac{z^{6}}{648}}$.

For heteronuclear translational relaxation, R_1_ takes the following expression [86,120]:(15)$$R_{1\mathrm{Diff}}^{\mathrm{ij}}\left(\omega_{i}\right)=\frac{A_{D}^{\mathrm{ij}}}{d_{\mathrm{ij}}D_{\mathrm{ij}}}\left[\tilde{J}\left(z(\omega_{i}-\omega_{j})\right)+3\tilde{J}\left(z(\omega_{i})\right)+6\tilde{J}\left(z(\omega_{i}+\omega_{j})\right)\right].$$

Here $A_{D}^{\mathrm{ij}}=\frac{1}{9}\gamma_{i}^{2}\gamma_{j}^{2}{\text{ћ}}^{2}{\left(\frac{\mu_{0}}{4\pi }\right)}^{2}n_{j},$ and the subscripts i and j refer to the two different nuclei. Equations (13) and (14) have been used successfully to describe the translational relaxation of ^1^H and ^19^F in the IL electrolyte composed of [BMIM][TFSI] with up to 1M LiTFSI at different temperatures in the range 303–333 K [62] and in the case of [P_4441_][TFSI] (tributyl methyl phosphonium bis(trifluoromethanesulfonyl)imide)) and [N_4441_][TFSI] (tributyl methyl ammonium bis(trifluoromethanesulfonyl)imide) with up to 0.6 molal concentration of LiTFSI at different temperatures in the range 298–338 K [110].

On the other hand, if isotropic molecular rotations with an average correlation time are assumed as a first approach, a Lorentzian function (L) can express the contribution of molecular rotations to the spin-lattice relaxation rate [86,87,88,89,90,91,92,93]:(16)$$R_{1\mathrm{rot}}^{L}\left(\omega \right)=A_{R}\left[\frac{\tau_{R}}{1+{\left(\omega \right)}^{2}\tau_{R}{}^{2}}+\frac{4\tau_{R}}{1+{\left(2\omega \right)}^{2}\tau_{R}{}^{2}}\right],\;$$
where $A_{R}=\frac{3}{10}\gamma^{4}{\text{ћ}}^{2}{\left(\frac{\mu_{0}}{4\pi }\right)}^{2}\frac{1}{r^{6}}\;$is the corresponding amplitude reflecting the strength of the relevant dipolar interactions, r is the effective distance between two nuclei located in the same molecule and τ_R_ is the average rotational correlation time. Even though this is the simplest model for rotational relaxation, it has been applied with great success to describe intramolecular relaxation in ILs and IL electrolytes [62,104,105,108,110,117,118]. Other works have considered more realistic models where the high viscosity of ILs, a distribution of motional correlation times and the molecular anisotropy of ions have been taken into account [103,106,121]. However, the mathematical expressions of these models are more complex, and more parameters need to be calculated. Specifically, a Cole–Davidson (CD) form [103,106,111] has been used to describe rotational dynamics:(17)$$R_{1\mathrm{rot}}^{\mathrm{CD}}\left(\omega \right)=A_{R}\left[\frac{\sin\left(\beta \arctan\left({\omega \tau }_{\mathrm{CD}}\right)\right)}{\omega {\left(1+{\left({\omega \tau }_{\mathrm{CD}}\right)}^{2}\right)}^{\beta /2}}+\frac{2\sin\left(\beta \arctan\left(2{\omega \tau }_{\mathrm{CD}}\right)\right)}{\omega {\left(1+{\left(2{\omega \tau }_{\mathrm{CD}}\right)}^{2}\right)}^{\beta /2}}\right].$$

Here the rotational correlation time is given by τ_R_ = βτ_CD_, where 0 < β ≤ 1 is a phenomenological stretching parameter and for β = 1 a Lorentzian form is obtained, and τ_CD_ is the average correlation time. In this model, the rotational motion is described as thermally activated jumps over potential barriers where β describes the spread of them, or in other words this parameter is connected to the existence of a distribution of correlation times.

For elongated ions which experience anisotropic reorientations, Woessner’s model (W) considers different correlation times for reorientations around the short (τ_S_) and long (τ_L_) molecular axes [122]. This model has been adopted to successfully describe the rotational relaxation of BMIM^+^ in BMIMCl IL [121] and [BMIM][BF_4_] ILs [123] and is given by the following equation:(18)$$R_{1\mathrm{rot}}^{W}\left(\omega \right)={\;A}_{R}\left[J^{W}\left(\omega \right)+4J^{W}\left(2\omega \right)\right],$$
where $J^{W}\left(\omega \right)=\frac{2}{15}\left[A_{0}\frac{2\tau_{1}}{1+{\left({\omega \tau }_{1}\right)}^{2}}+A_{1}\frac{2\tau_{2}}{1+{\left({\omega \tau }_{2}\right)}^{2}}+A_{2}\frac{2\tau_{3}}{1+{\left({\omega \tau }_{3}\right)}^{2}}\right]$; τ_1_ = τ_S_, τ_2_ = (1/τ_S_ + 1/τ_L_)^−1^ and τ_3_ = (1/τ_S_ + 4/τ_L_)^−1^; the parameters $A_{0}={\left(1-3\cos^{2}\alpha \right)}^{2}/4$, $A_{1}=3\sin^{2}\alpha /4$ and $A_{2}=3\sin^{4}\alpha /4$ are the geometric factors depending on the angle between the r vector (r is the intramolecular distance between two nuclei) and the long axis of the molecule, α. If translational and rotational motions are assumed to be statistically independent and/or dominant in different time scales, the total spin-lattice relaxation rate will be given by:(19)$$R_{1\mathrm{IL}}^{I\;=\;\text{½}}\left(\omega \right)=R_{1\mathrm{intra}}\left(\omega \right)+R_{1\mathrm{inter}}\left(\omega \right)=R_{1\mathrm{rot}}\left(\omega \right)+R_{1\mathrm{Diff}}^{\mathrm{ii}}\left(\omega \right)+R_{1\mathrm{Diff}}^{\mathrm{ij}}\left(\omega \right),$$
where homonuclear (ii) and heteronuclear (ij) translational relaxation have been considered. Equation (19) has been used to address the measured relaxation rate dispersions of nuclei with I = ½ in ILs in many works [62,103,104,105,106,108,110,111,117,118,121].

Most of the studies about ILs with lithium salts that use the FFC-NMR relaxometry technique have only focused on the relaxation of dipolar nuclei that belong to ions of the IL, but a few of them have analyzed the quadrupolar relaxation of ^7^Li [62]. The spin-lattice relaxation of nuclei with spin I ≥ 1, such as ^7^Li with I = 3/2, is mainly due to the interaction between the nuclear electric quadrupole, as characterized by its moment Q, and the fluctuating EFG tensor at the nuclear site caused by the surrounding intra and intermolecular charge distributions. For a simple Li^+^, the origin of the EFG is intermolecular, and its temporal fluctuations include both long-range translational diffusion and local (short-range) dynamics of Li^+^. In the case of [BMIM]TFSI with LiTFSI, the spin-lattice relaxation rate of lithium was explained by fluctuations of the EFG tensor mostly due to the local translational motion of Li^+^ within the coordination sphere surrounded by anions [62]. Assuming isotropic diffusional motion given by jumps with an average correlation time τ_jump_ as a first approach, a Lorentzian form can express the spin-lattice relaxation rate [62]:(20)$$R_{1\mathrm{IL}}^{\mathrm{Li}}\left(\omega \right)=A_{Q}\left[\frac{\tau_{\mathrm{jump}}}{1+{\left(\omega \right)}^{2}\tau_{\mathrm{jump}}{}^{2}}+\frac{4\tau_{\mathrm{jump}}}{1+{\left(2\omega \right)}^{2}\tau_{\mathrm{jump}}{}^{2}}\right],$$
where $A_{Q}=\frac{3\pi^{2}}{10}\frac{\left(2I+3\right)}{I^{2}\left(2I-1\right)}{\left(1+\frac{\eta^{3}}{3}\right)}^{2}C_{Q}^{2}$; τ_jump_ ≡ ⟨r⟩^2^/6D_s_, ⟨r⟩^2^ is the average one-jump distance and D_s_ is the diffusion coefficient; η is the asymmetry parameter of the EFG and C_Q_ is the quadrupolar coupling constant. If an axially symmetric EFG tensor is assumed, η is zero.

MDS is a helpful guide in identifying proper relaxation models since it allows for a determination of diffusion coefficients and rotational correlation times without any further model assumptions. Therefore, relaxometry experiments and MDS have been combined [117,118] to validate the assumptions made to derive the analytical expressions for the employed relaxation models in ILs. It was demonstrated that nuclear magnetic resonance properties calculated from MDS reproduce measured dispersion curves and temperature trends faithfully [117]. The good agreement between experiment and computation showed that the assumptions made by using, for instance, a Lorentzian model for rotational relaxation are applicable at least in a qualitative way if the ions do not present reorientational dynamics that are substantially anisotropic [118].

### 4.2. Deep Eutectic Solvents (DESs)

Although NMR spectroscopy is used in the studies of DESs [36,52,54], the use of relaxation times is in its infancy, with a few examples to highlight [36,124,125,126,127,128]. Despite this, the following works add to the importance of relaxation times as a valuable tool in ion dynamics studies.

Paterno et al. [36] studied the xAlCl_3_:amide DESs series with x ranging from 1 to 1.7 moles. The amides were of varying alkyl chain lengths and included: acetamide (CH_3_CONH_2_), propionomide (CH_3_CH_2_CONH_2_) and butyramide (CH_3_CH_2_CH_2_CONH_2_). The objectives of the study were to determine the aluminum ion species interactions and their corresponding dynamics and how both were affected by the amide type. Both ^1^H and ^27^Al T_1_ measurements were determined over the temperature range of 293–363 K. The ^1^H T_1_ data revealed differences in the local environments of the terminal species of each amide. At fixed molar ratios, the T_1_ values for the amide (NH_2_) group decreased with increasing alkyl chain length, which the authors surmised corresponded with the expected decrease in the dielectric screening ability of the amides. However, unlike the NH_2_ group, at fixed molar ratios the ^1^H T_1_ for the terminal CH_3_ group was longest and shortest for the propionamide and butyramide DESs, respectively. From this result the authors surmised a balancing of the flexibility associated with shorter alkyl chains, and the electrostatic screening afforded by the greater dielectric constants provided the most suitable local environment for the rotational motion of CH_3_ about its principal C_3V_ axis. Additional revelations about the DES local environments were revealed by the ^27^Al T_1_ data. Firstly, despite it generally having the smallest size and being the most symmetric of the identified aluminum ion species (AlCl_4_^−^, AlCl_2_(amide)^+^ and AlCl_2_(amide)_2_^+^) in the DESs, the AlCl_4_^−^ species experienced the largest EFG, as evidenced from it having the shortest T_1_ for all amides at every concentration and temperature. Activation energies determined from the Arrhenius plots of the variable temperature T_1_ data also increased for the AlCl_4_^−^ species but decreased for the free AlCl_3_ and AlCl_2_(amide)^+^ species with increasing molar ratios. The authors surmised that since the local electrostatic interactions would also be stronger for greater AlCl_3_ concentration, thereby restricting the species’ rotational and translational mobilities, there would be a need for greater activation energies. In contrast to the AlCl_4_^−^ species, the AlCl_2_(amide)^+^ species dynamics were very dependent on the flexibility of the amides’ alkyl chains, with their conformational flexibility effectively reducing the strength of the local EFG the AlCl_2_(amide)^+^ species experienced.

Alfurayj et al. [124] studied the effect of different water concentrations (0, 0.1, 1, 10 and 28.5 wt%) on the dynamics of the DES ethaline (a 1:2 mol% mixture of choline chloride and ethylene glycol) by using different experimental techniques, among them NMR diffusometry and broadband relaxometry, and molecular dynamics simulations. The authors showed that the translational and rotational motions of choline and ethylene glycol in ethaline accelerated with the presence of water, and the higher the water content, the faster these motions became. This was consistent with the decreasing viscosity when water was added. The ^1^H spin-lattice relaxation profiles of choline and ethylene glycol were described by combining the models given by Equations (14) and (16).

On the other hand, Fraenza et al. [126] used these same models to describe the ^1^H spin-lattice relaxation of choline and glycerol in the DES glyceline (a 1:2 mol% mixture of choline chloride and glycerol). The authors combined multiple NMR techniques such as FFC relaxometry, PFG diffusion and ^13^C NMR relaxation to study the dynamics of choline and glycerol as a function of choline chloride concentration (0, 5, 10, 25 and 33 mol%). They detected faster rotational and translational dynamics of all species in glyceline because of the disruption of the glycerol H-bonding network as choline chloride was added until the eutectic concentration (33 mol% choline chloride).

### 4.3. Organic Solvent Electrolytes

There are numerous relaxation time (T_1_, T_2_) studies on organic solvent electrolytes. Although Li^+^ electrolytes have featured most prominently compared to those of other batteries, there have been some studies where relaxation times have provided insight into ion–ion and ion–solvent interactions, as exist in multivalent ion electrolytes. The following selections highlight the variety of applications of relaxation times.

Peng et al. [129] used ^11^B T_1_ and T_2_ measurements to study the LiBF_4_ and LiPF_6_ salts in binary carbonate mixtures composed of EC and DMC. The LiPF_6_ salt concentrations ranged from 0.01 M to 1.2 M, while that for the LiBF_4_ were only 0.01 and 1 M. The EC/DMC solvent compositions used included 80/20, 50/50, 20/80 and 0/100 for both salts. Similar values were obtained for the T_1_ and T_2_ of the LiBF_4_ mixtures, indicating that the media were being investigated in the motional narrowing regime. At lower DMC content (ergo higher solvent polarity), the lower salt concentration electrolyte had significantly larger (>10×) T_1_ and T_2_ values compared to the 1 M electrolyte. However, with increasing DMC content the relaxation values for both electrolytes converged, indicating stronger interactions between the anions and cations. Larger ion–ion interactions were observed for the LiBF_4_ electrolytes compared to the LiPF_6_ ones, as evidenced by the decreasing ^11^B relaxation times. This effect was magnified by an increase in DMC concentration, which effectively reduced the overall solvent polarity, further enhancing the interactions.

Chen et al. [130] used ^19^F T_1_ and T_2_ measurements in their study of varying concentration of magnesium bis (trifluoromethanesulfonyl) imide (MgTFSI_2_) in a 1,2-dimethoxyethane (DME) solvent. This is a common Mg^2+^ battery electrolyte, but there are questions about its conduction properties. For example, the ionic conductivity reaches a maximum at 0.6 M but decreases until the solution is saturated. Additionally, computational studies suggest that MgTFSI_2_ in DME exists primarily as fully solvated free ions independent of concentration [131,132], which goes against the formation of ion pairs with increasing salt concentration. One argument put forth to explain the saturation behavior is the increase in permittivity caused by the re-dissociation process of contact-ion-pair (CIP) formation at higher concentrations [133]. This process is known to depend on the increase in the orientational polarizability of CIP species at higher concentrations, but the roles of the conformational flexibility of DME and the unbounded anions are undetermined. To elucidate the complexities of this system, the authors conducted a quantitative analysis of Mg^2+^ solvation characteristics with salt concentration as the determining variable. ^19^F T_1_ and T_2_ are shown in Figure 1 as functions of both salt concentrations (a and b) and temperatures (c and d), respectively. The decrease in T_1_ correlated with the increase observed in the solution viscosity (not shown) and was associated with a decrease in the rotational dynamics of the CF_3_ groups. The effect of viscosity is evidenced in the T_2_ data at higher concentrations (≥0.36 M). However, the smaller T_2_ values at lower salt concentrations (which corresponded to a drop in the self-diffusion coefficient) pointed to the formation of CIP Mg(TFSI)_m_(DME)_n_ (m, n = 1,2) structures and the corresponding restricted dynamics. With the aid of ^17^O spectra, the authors were able to confirm the structure of the CIP changes with increasing salt concentration, going from Mg(TFSI)(DME)^+^ between 0.01–0.05 M, Mg(TFSI)(DME)_2_^+^ at 0.06 M and [Mg(DME)_3_]^2+^ at higher concentrations.

Li-Sulfur (Li-S) battery electrolytes often suffer from the complexities of the oxidation and reduction Li-S chemistry and conduction pathways [134,135,136]. For example, in the conversion of S8 to Li_2_S, soluble intermediate lithium polysulfide species (Li_2_S_n_, n ≥ 4) are formed and dissolve into the organic electrolyte, resulting in poor Coulombic efficiency and particularly in severe loss of capacity by reacting with the Li electrode [137]. One method used to combat the polysulfide dissolution problem is the implementation of high Li salt concentrations. In these systems, the increase in Li salt concentration increases the CIP formation whereby Li coordination is accomplished by both the salt anion and the solvent molecules. Exceeding the concentration level whereby most solvent molecules are coordinated to Li ions leaves few ‘free’ solvent molecules available to solubilize the polysulfide species. One system which has been studied in recent years [138,139,140,141,142] comprises the LiTFSI salt in acetonitrile (MeCN) to form the (MeCN)_2_-LiTFSI solvate electrolyte. This electrolyte suffers from lower ionic conductivity and correspondingly high viscosity, both of which can be improved using cosolvents such as hydrofluoroether (HFE). Shin et al. [143] studied the effect of the HFE structure (see Figure 2) on the (MeCN)_2_-LiTFSI solvate electrolyte using ^7^Li T_1_ measurements. Values were determined between 238–343 K for the neat and HFE analogues of the (MeCN)_2_-LiTFSI solvate electrolytes in volume ratios of 2:1, 1:1 and 1:2.

As shown in Figure 3, with increasing HFE content the T_1_ minimum shifted to lower temperatures irrespective of the HFE type. This behavior indicates enhancement in the Li^+^ mobility caused by the reduced viscosity. The order of the T_1_ minimum followed the inverse of the analogue’s viscosity. The T_1_ minimum also shifted to faster times, indicating a more efficient quadrupolar relaxation mechanism, which the authors surmised was the result of increased asymmetry possibly due to the replacement of coordinated MeCN by HFE molecules, which have high electron density around the oxygen and fluorine atoms. The fitting of the T_1_ data to the BPP model for quadrupolar nuclei (Equation (21)) allowed for the extraction of τ_c_ and ω_q_.
(21)$$\frac{1}{T_{1}}=\frac{\omega_{q}^{2}}{50}\left(\frac{\tau_{c}}{1+{\;\omega }_{o}^{2}\tau_{c}^{2}}+\frac{4\tau_{c}}{1+4\omega_{o}^{2}\tau_{c}^{2}}\right).$$

Here ω_q_ is the quadrupolar coupling constant (QCC), and it was found to increase with increasing HFE, suggesting the presence of larger oscillating electric field gradients (EFGs) around the Li ions. Of the HFEs, the TTE and OTE (see Figure 2) both had larger ω_q_ values which correlated well with their shorter T_1_. This result, combined with them having the lower τ_c_ values, supported TTE and OTE, inducing greater asymmetry compared to BTFE and ETE.

### 4.4. Confined Liquid Electrolytes

FFC-NMR relaxometry studies about confined liquid electrolytes have been mostly concentrated on ILs without [105,108,121,123,144,145,146,147] and with lithium salt [62], and a few of them have been concentrated on aqueous electrolyte solutions [148]. Most of them have been focused on the dynamics of the cation (^1^H NMR) [121,123,144,145,147,148] and only a few of them have studied both cation and anion dynamics (^1^H and ^19^F NMR) [62,105,108,110,146]. Many of these works have shown that the confinement causes a slight slowing-down of the ion dynamics, which is consistent with the fact that the ionic conductivity of the confined electrolyte is usually smaller compared to the corresponding value for the pure IL [121]. However, the opposite effect, enhanced ionic conductivity, could be observed depending on the type of confining matrix that is used and how it interacts with ions, since this interaction might prohibit the formation of ion clusters or associates [123,144,148]. For instance, in the following cases enhanced ionic conductivity has been observed: (i) the gel electrolyte made of an aqueous solution of tetramethylammonium bromide (TMABr) confined into the low molecular weight organic gelator methyl-4,6-O-(p-nitrobenzylidene)-α-D-glucopyranoside [144,148]; (ii) the gel polymer electrolyte based on ethoxylated bisphenol A dimethacrylate and 1-butyl-3-methylimidazolium tetrafluoroborate ([BMIM][BF_4_]) IL [123]. Independent of the type of IL and confining matrix, all these works [62,105,108,110,121,123,144,145,146,148] have shown that in the case of the bulk IL electrolyte there is no frequency dependence in the low-frequency window (in general below 1 MHz) of the spin-lattice relaxation rate profiles. In contrast, for the confined electrolyte a well-pronounced dispersion is seen in this frequency range, which is a fingerprint of the interactions between the ions and the confining material.

#### 4.4.1. Ionogels (IGs)

The molecular dynamics become quite complex when an IL is confined in a mesoporous inorganic matrix as it occurs in an IG. The dynamic properties of ILs in confinement strongly depend on the relative size of the ions and the pores and on the specific IL and matrix composition. Therefore, two different approaches have been considered to describe the spin-lattice relaxation dispersions of IGs, namely, the two and one ion population approaches.

The two-population approach considers that the size of the pores or cavities of the matrix is big enough to allow the distinction of two ion populations, namely, a core fraction near the pore center and a surface fraction near the confining walls [105,145,148]. The first one is dominant, and its ions dynamically behave as “bulk-like”. For the second one, the ions interact with the pore walls, and therefore their dynamics are affected by the matrix surface topology. Other work has identified the bulk-like phase as ions at least a few ionic radii distant from the matrix surfaces and the surface-like phase as those ions close to the matrix surfaces [62]. Then, the total relaxation rate is given by:(22)$$R_{1\mathrm{IG}}^{2\;\mathrm{populations}}\left(\omega \right)=R_{1}^{\mathrm{core}}\left(\omega \right)+R_{1}^{\mathrm{surf}}\left(\omega \right).$$

Here it is assumed that the residence lifetime of the ions on the surface is longer than the translational correlation time in bulk. In this way, the exchange between the two fractions is slow enough to allow us to distinguish between these two populations, and the contribution to the relaxation from the exchange process can be omitted. Furthermore, weight factors for the two fractions of the liquid can be expressed explicitly by multiplying each term of Equation (22) [62]. The spin-lattice relaxation rate of ions that belong to the bulk-like phase (R_1_^core^) is assumed to be given by the Equation (19) or (20), depending on the value of the quantum number I of the nuclei [62,105,145,148]. On the other hand, the relaxation of ions interacting with the confining walls (R_1_^surf^) has been described in two different manners. One of them considers that their dynamics can be represented by the same models of Equation (19) or (20) but with a different diffusion constant and rotational correlation time [105,145]. In other words:(23)$$R_{1\mathrm{IG}}^{2\;\mathrm{populations}}=R_{1}^{\mathrm{core}}\left(\omega \right)+R_{1}^{\mathrm{surf}}\left(\omega \right)=R_{1\mathrm{Diff}}^{\mathrm{core}}\left(\omega \right)+R_{1\mathrm{Rot}}^{\mathrm{core}}\left(\omega \right)+R_{1\mathrm{Diff}}^{\mathrm{surf}}\left(\omega \right)+R_{1\mathrm{Rot}}^{\mathrm{surf}}\left(\omega \right)$$

Here four different dynamic parameters are obtained: D^core^, D^surf^, τ_R_^core^ and τ_R_^surf^. The other manner assumes that the translational displacements of ions become restricted due to the interaction with the confining material, taking different local orientations determined by the topology of the cavity surface. These molecular reorientations mediated by translational displacements (RMTD) of the molecules, which occur on a much slower time scale than bulk rotational or translational diffusion, have been considered as the main relaxation mechanism for ions that interact with the confining walls [62,148]. It is important to notice that the RMTD model can be taken under consideration when the investigated system consists of a polar surface and polar solvents. Then, in this case the total relaxation rate is given by:(24)$$R_{1\mathrm{IG}}^{2\;\mathrm{populations}}=R_{1}^{\mathrm{core}}\left(\omega \right)+R_{1}^{\mathrm{surf}}\left(\omega \right)=R_{1\mathrm{Diff}}^{\mathrm{core}}\left(\omega \right)+R_{1\mathrm{Rot}}^{\mathrm{core}}\left(\omega \right)+R_{1\mathrm{RMTD}}^{\mathrm{surf}}\left(\omega \right)$$
where:(25)$$R_{1\mathrm{RMTD}}^{\mathrm{surf}}\left(\omega \right)={\;A}_{\mathrm{RMTD}}\left[J^{\mathrm{RMTD}}\left(\omega \right)+4J^{\mathrm{RMTD}}\left(2\omega \right)\right]$$

Here A_RMTD_ is a constant that depends on the type of nuclear interaction (dipolar or quadrupolar), on the microstructural features of the confining matrix, on the diffusion coefficient and on the fraction of molecules at the surface. J^RMTD^ can be expressed in terms of modes with wave numbers k as follows [149]:(26)$$J^{\mathrm{RMTD}}\left(\omega \right)=\frac{1}{{\left(2\pi \right)}^{2}}\int_{0}^{\infty }S\left(k\right)\frac{2\tau_{k}}{1+{\left({\omega \tau }_{k}\right)}^{2}}\mathrm{dk}.$$

S(k) is the orientational structure factor, which represents the geometrical information of the surface in terms of surface modes with wave number k, and τ_k_ = τ_k_(k). Both can take different expressions depending on the assumptions that are made. The motion of ions can be described by two-dimensional diffusion and represented by a Gaussian or Cauchy probability density depending on whether the adsorption is weak or strong, respectively [149]. This determines the expression for τ_k_. In general, a Gaussian probability density is assumed and in the simplest case S(k) is considered as a constant [62,148], which is equivalent to the assumption of an equipartition of the wave numbers describing different diffusion modes. In this case, $R_{1\mathrm{RMTD}}^{\mathrm{surf}}\left(\omega \right)$ is given by:(27)$$R_{1\mathrm{RMTD}}^{\mathrm{surf}}\left(\omega \right)=A_{\mathrm{RMTD}}\left[\omega^{-0.5}\left(f\left(\frac{\omega_{\max}}{\omega }\right)-f\left(\frac{\omega_{\min}}{\omega }\right)\right)+4{\left(2\omega \right)}^{-0.5}\left(f\left(\frac{\omega_{\max}}{2\omega }\right)-f\left(\frac{\omega_{\min}}{2\omega }\right)\right)\right].\;$$

Here $f\left(\omega \right)=\arctan\left({\left(2\omega \right)}^{0.5}+1\right)+\arctan\left({\left(2\omega \right)}^{0.5}-1\right)-\mathrm{arctanh}\left(\frac{{\left(2\omega \right)}^{0.5}}{\omega +1}\right)$. $\omega_{\min}=\frac{4D_{\mathrm{surf}}}{l_{\max}^{2}}\;$and $\omega_{\max}=\frac{4D_{\mathrm{surf}}}{l_{\min}^{2}}\;$represent, respectively, the minimum and maximum frequencies of diffusion modes on the matrix surface; D_surf_ denotes the surface diffusion coefficient, l_max_ describes the largest and l_min_ describes the smallest displacement distance along the surface. The two-population approach using Equations (24) and (27) has been used to successfully describe the spin-lattice relaxation profiles of hydrogen and fluorine in IL electrolytes made of [BMIM][TFSI] with the lithium salt LiTFSI confined in a silica matrix [62]. On the other hand, this model was also applied to describe ^1^H relaxation of an aqueous solution of tetramethylammonium bromide (TMABr) confined in a gel matrix made of a low molecular weight gelator [148].

Another common assumption for S(k) is that it follows a power law; therefore $R_{1\mathrm{RMTD}}^{\mathrm{surf}}\left(\omega \right)$ is given by [149]:(28)$$R_{1\mathrm{RMTD}}^{\mathrm{surf}}\left(\omega \right)=\tilde{A_{\mathrm{RMTD}}}\omega^{-p},\;0\;<\;p\;<\;1$$

The amplitude $\tilde{A_{\mathrm{RMTD}}}$ can be written as C × D^p−1^, where C is a constant that depends on the type of interaction (dipolar or quadrupolar), on the exponent p and on the type of probability density (Gaussian or Cauchy). The temperature dependence is in this case introduced by the diffusion coefficient D. The two-population approach using Equations (20) and (28) was used to depict beyond all expectation ^7^Li relaxation in IGs made of the IL [BMIM][TFSI] with LiTFSI confined in a SiO_2_ matrix [62]. This is a clear indication that RMTD can also be an efficient relaxation mechanism for quadrupolar nuclei such as ^7^Li.

On the other hand, the one-population approach considers that all ions interact with the confining walls all the time due to the small dimensions of cavities, and consequently it is not possible to distinguish two different populations of ions. Under this assumption, the model presented by Equation (28) has been used to describe the ^1^H and ^19^F spin-lattice relaxation profiles of the IL [BMIM][TFSI] confined in a porous glass [146]. In contrast, the ^1^H and ^19^F spin-lattice relaxation profiles of the IL [EMIM][FSI] (1-ethyl-3-methylimidazolium bis(fluorosulfonyl)imide) into a mesoporous MCM41 silica matrix have been explained by the two-dimensional (2D) translational diffusion of the ions in the vicinity of the confining walls and rotational motion [108]. The 2D diffusion was considered as a sequence of loops near the confining walls interrupted by time periods during which the ions were attached to the surface. Additionally, the rotational motion of the confined ions was described by two correlation times that were attributed to the possible anisotropic reorientation of these species. Therefore, the total relaxation rate in this case was expressed by [108]:(29)$$R_{1\mathrm{IG}}^{1\;\mathrm{population}}=R_{1}^{2D\;\mathrm{Diff}}\left(\omega \right)+R_{1}^{\mathrm{Rot},\;s}\left(\omega \right)+R_{1}^{\mathrm{Rot},\;f}\left(\omega \right)+R_{1}^{\mathrm{FM}}.$$

Here $R_{1}^{2D\;\mathrm{Diff}}\left(\omega \right)=A_{\mathrm{trans}}\tau_{\mathrm{trans}}\left[\ln\left(1+\frac{1}{{\left({\omega \tau }_{\mathrm{trans}}\right)}^{2}}\right)+4\ln\left(1+\frac{1}{{\left(2{\omega \tau }_{\mathrm{trans}}\right)}^{2}}\right)\right]$; $R_{1}^{\mathrm{Rot},\;l}\left(\omega \right)$ follows a Lorentzian form (Equation (16)) and the superscripts l = s, f stand for slow and fast rotations. Equation (29) includes a frequency independent term, $R_{1}^{\mathrm{FM}}$, which is associated with dynamical processes being too fast to lead to a visible dependence of the corresponding relaxation contribution on the resonance frequency. It might originate from the internal dynamics of the confined ions.

#### 4.4.2. Gel Polymers Electrolytes (GPEs)

Richardson et al. [65] used T_2_ measurements to illustrate the multiphase complexity of the poly (vinylidene fluoride) (PVDF)/PC/LiBF_4_ polymer gel electrolytes composed of 20 and 30% PVDF. The ^1^H T_2_ results revealed a multiphase system comprising a crystalline PVDF lamellae, an inter lamellae amorphous PVDF, a solvated amorphous PVDF and a liquid electrolyte. Due to the multiphase nature of the electrolytes, a single exponential fitting function as shown in Equation (5) was not applicable. Instead, the authors used a modified fitting function composed of the series addition of multiple single exponentials, as shown in Equation (30):(30)Mx,y=|Mo1| e−tT21+|Mo2| e−tT22+|Mo3| e−tT23, 
where $M_{o}^{1},{\;M}_{o}^{2}{\;\mathrm{and}\;M}_{o}^{3}$ represent the relative intensity of each phase, and $T_{2}^{1},{\;T}_{2}^{2}{\;\mathrm{and}\;T}_{2}^{3}$ are the respective phase’s spin–spin relaxation time. Due to the shortness of the T_2_ expected for the crystalline PVDF lamellae (microseconds), the authors surmised a three-phase fit was satisfactory for both the 20 and 30% PVDF electrolyte compositions. The intensities determined for the inter lamellae amorphous PVDF, solvated amorphous PVDF and liquid electrolyte phases were 12.7%, 35.7% and 51.6%, and 14%, 42% and 44% for the 20% and 30% PVDF electrolyte compositions, respectively. These results were supported by the ^1^H self-diffusion coefficients obtained by standard pulsed field gradient (PFG) measurements.

In the same way as IGs, when the IL is mixed with a polymer gel, the molecular dynamics of the ions become more complex because of the confinement that they experience. They interact with the polymer chains and therefore their translational displacements become restricted, taking different local orientations determined by the topology of the polymer network. In recent years, a new kind of GPE based on the entrapment of ILs in poly (IL) (PIL) hosts has emerged. One of the main benefits of this type of electrolyte is the prevention of phase separation (sweating out effect) and leakage of the liquid component during application. Recently, a new class of GPE [150,151,152,153,154,155,156,157,158] composed of a polymer, a salt and an IL has gained attention due to its higher ionic conductivity when compared to its gel counterpart [154,155]. Here we discuss a few of these studies.

In an earlier investigation of PIL systems, Gouverneur et al. [153] used ^7^Li T_1_ measurements to study the interactions between the polymer–solvent mixture and the Li^+^ ions with a specific focus on how the polymer concentration affects dynamics. The PILs studied were composed of 1-butyl-1-methylpyrrolidinium bis(trifluoromethanesulfonyl) imide (Pyr_14_TFSI), lithium TFSI and poly(diallyldimethylammonium) TFSI (PDADMATFSI) (see Figure 4a). Samples were composed of the 20:X:02 and 20:X:04 molar ratios series of IL:PIL:LiTFSI.

With increasing PIL concentration, the R_1_ maximum moves to higher temperatures for both salt concentrations (see Figure 4b,c), signifying decreasing Li^+^ dynamics. For each salt concentration, the E_A_ values decreased with increasing PIL content despite the reduction in long-range dynamics, as evidenced from both the ionic conductivity and self-diffusion measurements. The authors surmised that the decreasing E_A_ values may be due to the PIL’s positively charged backbone, thus explaining its ability to break up anion clusters that generally coordinate with Li^+^. The extracted $\tau_{c}$ values increased with PIL content, suggesting that despite the PIL’s aid in breaking up anion clusters, it impedes Li^+^ dynamics far more.

On the other side, Brinkkötter et al. [159] investigated the influence of the polymer on the local dynamics and the local environment of the lithium ion in the same ternary GPE studied by Gouverneur et al. [153], using frequency- and temperature-dependent ^7^Li T_1_ measurements. Even though they obtained the same results as Gouverneur et al., they discovered that the Cole–Davidson motional model (Equation (17)), with an Arrhenius dependence of the correlation time and a temperature-dependent quadrupole coupling constant, described the relaxation rates better than the BPP model did. Additionally, they made a comparison with the GPE based on PEO, and although the local Li^+^ motion is reduced by the presence of either polymer, the reduction is less effective in the PDADMA^+^ samples. Thus, they concluded that the mechanical stabilization of a liquid electrolyte by a polymer can be achieved at a lesser drawback of Li^+^ motion when a cationic polymer is used instead of the neutral PEO because the first one induces a fragmentation of lithium-anion clusters, whereas the neutral polymer interacts via a coordination state to the Li^+^.

In a later study, Brinkkötter et al. [160] used ^7^Li T_1_ measurements in their determination of the effect of the poly (ionic liquid) (PIL) cationic structure on the physical properties of ternary GPEs. Like Gouverneur et al. [153], the IL and salt used were Pyr_14_TFSA and LiTFSA. The PILs used are shown in Figure 5a. The molar ratio of the PIL:IL:LiTFSA system was 4:20:4. In spite of the heterogeneous nature of the materials, all the T_1_ data required only a single exponential fitting equation, which the authors surmised was due to either the detection of an average value due to the fast exchange between the various components or the fact that the difference between the various T_1_ values were within the experimental error of the measurements. Over the temperature range (298–423 K) studied, the R_1_ (1/T_1_) data exhibited a maximum that was sample-dependent with that for PIL_1ter_ and PIL_2ter_ (see Figure 5b) occurring at the lowest and highest temperatures, respectively. The authors used the BPP model to extract the $\tau_{c}$ and the activation energies (E_A_) from the Arrhenius fits of the respective T_1_ data using the equation:(31)$$\tau_{c}={\;\tau }_{0}\exp\left(\frac{-E_{A}}{\mathrm{RT}}\right).$$

Here τ_o_ is the pre-factor, R is the gas constant and T is the temperature in Kelvin. Contrary to the R_1_ maximum location for PIL_1ter_, which suggested that it had the fastest local ion dynamics of all the composites, it also had the highest *E_A_* value, whereas the PIL_2-4,ter_ all had the same value. From this, the authors surmised that the PIL_1ter_ system had localized connected domains of Pyr_14_TFSA/LiTFSA which were responsible for the enhanced dynamics despite PIL_1_ having a higher glass transition temperature and the lowest ionic conductivity.

FFC-NMR relaxometry has also been used to study GPEs, where the confined liquid electrolyte is based on ILs [110,121,123]. In the same way as IGs, the dynamic properties of ILs in confinement strongly depend on the relative size of the ions and the cavities and on the specific IL and polymer composition. Therefore, the two different previously discussed approaches considered for IGs have been utilized as well to describe the spin-lattice relaxation rate dispersions of GPEs (the two and one ion population approaches).

Rachocki et al. [123] studied BMIM^+^ dynamics in the GPE made of [BMIM][BF_4_] and bisphenol A ethoxylate dimethacrylate (bisAEA4) at the mass ratio of 20:80 and in the range of temperatures 248–343K. The relaxation of the cation was successfully explained by using the two-population approach given by the model of Equation (24). The “bulk-like” fraction was analyzed in terms of translational diffusion using Torrey’s model [119] and local molecular reorientations around a long and a short molecular axis of the IL cation using Woessner’s model (Equation (18)). On the other hand, the fraction of cations interacting with the polymer was described by the RMTD model given by Equation (27). Additionally, the authors measured the ionic conductivity through electrochemical impedance spectroscopy, and they observed that its value for GPE was twice as high than that of pure IL. This effect agreed with the larger diffusion coefficients obtained from the analysis of the relaxation profiles for the ions interacting with the polymer compared to the corresponding value for the bulk ions. This conductivity enhancement was explained by the interaction of IL aggregates at the polymer/IL interface. It was speculated that this interaction leads to a decrease in the size of ion complexes and in the tendency to form ion-pairs or aggregates in the interphase, allowing ions to create new ionic-conducting pathways which can play an important role in ionic conduction observed at the microscopic level. At the same time, BMIM^+^ dynamics in the GPE made of BMIMCl and cellulose (Cell) at 10 wt% in the range of temperatures 298–368 K was studied by Kaszyńska et al. [121]. The model used to describe the ^1^H spin-lattice relaxation rate profiles was the same used by Rachocki et al. [123]. The conductivity measurements performed for pure IL and that confined in the Cell/BMIMCl GPE showed that the gelation only resulted in a small decrease of the ionic conductivity. This is clear evidence that the type of anion and polymer determine the final properties of GPE because, although both works used the same cation [121,123], the gelation effect on conductivity was different in each case.

Garaga et al. [110] investigated the cation and anion dynamics in two different GPEs made of the ILs [P_4441_][TFSI] or [N_4441_][TFSI], both with up to 0.6 molal concentration of LiTFSI and the polymethyl methacrylate (PMMA) polymer at 10 wt%. They measured ^1^H and ^19^F spin-lattice relaxation rate profiles at different temperatures, which were modeled by considering the one-population approach. RMTD was assumed as the main relaxation mechanism for hydrogen and fluorine in the polymer gels (Equation (27)); however, a Lorentzian term (Equation (16)) was added in the case of cation relaxation to consider molecular reorientations that they could experience when they are not interacting with the polymer. This was clear evidence that cations had a less restricted motion than anions under the presence of the polymer which was connected to the more probable interaction between anions and the polymer. In general, the confinement showed a small slow-down in the dynamics of ions.

## 5. Solid Electrolytes

The study of relaxation times in solid materials is generally more challenging compared to liquids. This is because of a number of factors including large quadrupolar coupling, low natural abundance or low resonance frequencies, all of which can affect the data determined, the time required for the measurements and the data analysis. In spite of these, relaxation times have been used in numerous studies on solid electrolyte materials and the following selections highlight their importance.

As previously stated, PEO is one of the most studied polymers. However, it suffers from low ionic conductivity and mechanical integrity. Several methods have been used to help offset these disadvantages, one of which has been the inclusion of inorganic fillers such as TiO_2_ [161], ceramics [162,163] and lithium-ion-conducting ceramic particles such as Li_1_._3_Al_0_._3_Ti_1_._7_(PO_4_)_3_ (LATP) [162,163,164,165]. In one such study, Peng et al. [166] studied the effect of the doped LATP filler on the polymer electrolyte (PE) composed of PEO and the lithium trifluoromethanesulfonate salt (LiTf, LiCF_3_SO_3_). The ceramic polymer electrolyte (CPE) contained 55% of the doped LATP (labeled LICGC, Li_2_O-Al_2_O_3_-SiO_2_-P_2_O_5_-TiO_2_-GeO_2_) ceramic and both ^7^Li and ^19^F T_1_ were determined. These results were compared with the ceramic-free polymer electrolyte to ascertain the ceramic’s effect on ion dynamics and the interfacial interactions in the polymer–ceramic composite electrolyte. Three Li components were identified in the CPE: the ceramic component, the immobile PE component, and the mobile PE component. Because of the difficulty in resolving the signals, the authors used a biexponential fit for the relaxation profile to determine the values for the ceramic and mobile CPE components:(32)$$I_{\left(\tau \right)}={\;I}_{\left(o\right)}\left[f\left(1-2\exp\left(\frac{-\tau }{T_{1}^{\mathrm{LICGC}}}\right)\right)+\left(1-f\right)\;\left(1-2\exp\left(\frac{-\tau }{T_{1}^{\mathrm{mobile}\;\mathrm{Li}\;\mathrm{in}\;\mathrm{PEO}}}\right)\right)\right],$$
where $I_{\left(\tau \right)}$ is the amplitude of the central peak of the ^7^Li spectra at τ, f is the fraction of Li, $T_{1}^{\mathrm{LICGC}}$ represents the T_1_ of the ceramic and $T_{1}^{\mathrm{mobile}\;\mathrm{Li}\;\mathrm{in}\;\mathrm{PEO}}$ is the value for the mobile Li in PEO. The immobile component of PE was analyzed from the satellites of the ^7^Li spectra. As shown in Figure 6a, the ceramic hinders both the immobile and mobile Li components, as evidenced by the reduced T_1_ values of the CPE compared to the PE. It is noteworthy that the ^7^Li T_1_ values for the LICGC were pristine and in CPE were identical, indicating that the quadrupolar environment in both were similar. The ^19^F data for all species showed T_1_ minimum (see Figure 6b). That for the mobile F occurred around 40 °C, which was lower than that for the mobile Li. The authors attributed this difference to the fast rotation of the CF_3_ group around the C-S bond. As expected, the T_1_ for the mobile F was shorter in the CPE than in the PE. Irrespective of the reduced mobility of both ions, the ceramic’s presence in the PE caused an increase in the Li^+^ transference number, indicating greater immobilization for anions than cations.

The similarities between Li and sodium (Na) ions have contributed to the synthesis and investigation of Na ion electrolytes such as Li ones [167,168,169,170,171]. Pope et al. [171] used ^23^Na spectra and T_1_ to study SPE based on the poly (2-acrylamido-2-methyl-1-propane-sulfonic acid) (PAMPS) monomer, Na^+^ and dimethylbuttylmethoxyethyl ammonium (N_114(201)_) cations in the 2:8 molar ratio (see Figure 7a,b). The authors’ objectives included a direct comparison of the Na^+^ ion transport with that of Li^+^ in the lithium analogue electrolyte and the determination of the ether group’s role in the Na^+^ solvation and subsequent dissociation. As shown in Figure 7c, two distinct environments were discerned from the bi-exponential fitting of the T_1_ data. The shorter T_1_ component was assigned to the mobile Na^+^ cations while the longer T_1_ component was due to the immobile or bound Na^+^ cations associated with the sulfonate groups. Evidence of increasing segmental dynamics is visible from the dramatic decrease in values of the immobile Na^+^ cations above the glass transition temperature (T_g_) with increasing temperature, especially between 323–333 K.

One of the most studied solid electrolytes is the NASICON (sodium (Na) Super Ionic CONductor) family [172,173,174,175,176] with structure Na_3+x_Sc_2_(SiO_4_)_x_(PO4)_3-x_ (see Figure 8). Like most solid electrolytes, this family suffers from low room temperature ionic conductivity. The highest conductivity value observed at 298 K is 6.9 × 10^−4^ S/cm, obtained for the x = 4 electrolyte [177]. Zinkevich et al. [176] used multi-nuclear (^23^Na, ^31^P and ^45^Sc) spectra and relaxometry studies to investigate the sodium ion dynamics at the microscopic scale. The temperature-dependent ^23^Na R_1_ (1/T_1_) data for x = 0.1–0.8 all showed a maximum that occurred at higher temperatures with increasing x. Due to the difference in slopes between the high and low temperature sides of each BPP modeled v-shaped curve, the authors surmised the existence of correlated motions.

The authors used the Arrhenius relationship for a thermally activated process (Equation (33)) to determine the E_A_ values.
(33)$$\tau_{c}\left(T\right)={\;\tau }_{298\;K}\exp\left(\frac{E_{A}}{k_{B}}\;\left(\frac{1}{T}-\frac{1}{298\;K}\right)\right).$$

Here $\tau_{298\;K}\;$is the correlation time at room temperature, and T is the temperature. The E_A_ values determined decreased with increasing x, ergo increasing Si content. The authors attributed this effect to the expansion of the unit cell volume, which allows the transport of Na^+^. However, this process is also impeded by the increasing concentration of Na^+^, which reduces the vacancy number, thereby lowering the ionic conductivity.

Lithium- and sodium-doped solid glass electrolytes have gained attention for solid-state applications [178,179,180,181,182]. This is because of the decoupling of the ionic conduction from the media’s structural relaxation and the possibility of its tuning to provide the favored properties. One method of tuning is through the combination of multiple glass formers. These mixed-glass formers oftentimes experience non-linear changes in the materials’ electrical, thermal, mechanical, and chemical properties. Both positive and negative non-linear changes are possible and are referred to as the mixed-glass former effect (MGFE). The dynamic characterization of these materials’ by NMR spectra and T_1_ measurements has been invaluable by providing correlation times and activation energies that differ from those obtained by dc conductivity measurements. These deviations have been instrumental in the realization of the decoupling between the ionic conduction and structural relaxation pathways. The following results are examples of the invaluable information gleaned from T_1_ determination.

Storek et al. [180] used ^23^Na spectra and T_1_ measurements in their comparison of ion dynamics in two glasses—0.33Na_2_O + 0.67[xB_2_O_3_ + (1 − x)2SiO_2_] with a known negative MGFE and 0.35Na_2_O + 0.65[xB_2_O_3_ + (1 − x)P_2_O_5_] with a known positive MGFE. In the case of the negative MGFE electrolyte, the R_1_ (1/T_1_) results decreased with increasing xB_2_O_3_ concentration and displayed a minimum between 0.6–0.8, which is indicative of slow ion dynamics. By comparison, the positive MGFE electrolyte showed an increasing R_1_ with a maximum at x ≅ 0.4 over the temperature range studied. Activation energies determined were larger than those from both dc conductivity and tracer diffusion measurements and were explained using a percolation model.

In their study of the ternary glass family of electrolytes of the form yNa_2_S + (1 − y)[xSiS_2_ + (1 − x)PS_5/2_], Shastri et al. [182] used ^23^Na T_1_ to determine the distribution of activation energies around the sodium ions due to the heterogeneous nature of the materials. These distributions correlated with a mean barrier height for the sodium ion hopping motion facilitated by the dominant quadrupolar interaction. Unlike the Storek study [180], no pronounced MGFE was observed; however, the activation energy barrier height decreased with increasing SiS_2_ concentration. Additionally, the authors determined through their use of the Anderson–Stuart model [179,183] that the activation energy was the sum of the Coulombic contribution due to the mobile sodium ions interacting with other ions in the glass and a strain contribution caused by the cations’ displacement of other ions during their motions. They also determined that the energy barriers were strongly dependent on the materials’ dielectric constants.

The use of FFC-NMR relaxometry to study solid electrolytes is quite scarce basically because of two reasons. On one side, the short value of the T_2_ makes signal measurement unfeasible, and on the other side, even when it is possible to measure the signal, the values of T_1_ at low magnetic fields in general are shorter or in the same range than the switching time (2–3 ms). This makes T_1_ measurement unachievable due to electronic limitations. However, a few works using FFC-NMR to investigate solid electrolytes, such as glasses [178], glass ceramics [184] and polymer electrolytes [185], have been reported.

Gabriel et al. [178], used ^7^Li FFC-NMR to study the ionic jump motion and to determine the distribution of activation energies in the ternary 0.5Li_2_S + 0.5[(1 − x)GeS_2_ + xGeO_2_] glass system with x = 0 (reference) and 0.1 (higher conductivity) at different temperatures in the range of 200–450 K. Two models were employed and compared to explain the ^7^Li T_1_ rate profiles, namely, the multi-exponential (ME) autocorrelation function model and the power-law waiting time model. For the first one, the relaxation rate was described by a superposition of Lorentzian functions with a distribution of the correlation times, C(τ_c_), as follows:(34)$$R_{1\mathrm{glass}\_\mathrm{ME}}^{\mathrm{Li}}\left(\omega \right)=A_{Q}\left[\int_{0}^{\infty }C\left(\tau_{c}\right)\frac{\tau_{c}}{\left(1+\omega^{2}\tau_{c}^{2}\right)}{d\tau }_{c}+\int_{0}^{\infty }C\left(\tau_{c}\right)\frac{4\tau_{c}}{\left(1+{\left(2\omega \right)}^{2}\tau_{c}^{2}\right)}{d\tau }_{c}\right]$$

Here C(τ_c_) was specified by assuming thermally activated ion hops that follow an Arrhenius relation and a temperature-independent Gaussian distribution g(E_A_) for the activation energy:(35)$$g\left(E_{A}\right)=\frac{1}{\sigma \sqrt{2\pi }}e^{-\frac{{\left(E_{A}-\langle E_{A}\rangle \right)}^{2}}{2\sigma^{2}}}.$$

This model described the data very well for temperatures higher than 300 K, which indicated a change in the ^7^Li T_1_ mechanism upon cooling. On the other hand, to take into account other origins of the non-exponential relaxation, the power-law waiting time model was considered. It comes from the theory of a random walk in a random potential energy landscape and considers a random variation of barrier heights on the length scale of individual ion positions in the glass network [186,187]. Thus, this model leads to a spectral density of the Cole–Cole (CC) type and the relaxation rate in this case is given by:(36)$$R_{1\mathrm{glass}\_\mathrm{CC}}^{\mathrm{Li}}\left(\omega \right)=A_{Q}\left[\frac{4\omega^{\alpha -1}\tau_{\mathrm{CC}}^{\alpha }\sin\left(\alpha \pi /2\right)}{4+4{\left({\omega \tau }_{\mathrm{CC}}\right)}^{\alpha }\cos\left(\frac{\alpha \pi }{2}\right)+{\left({\omega \tau }_{\mathrm{CC}}\right)}^{2\alpha }}+\frac{16\omega^{\alpha -1}\tau_{\mathrm{CC}}^{\alpha }\sin\left(\alpha \pi /2\right)}{4+4{\left(2{\omega \tau }_{\mathrm{CC}}\right)}^{\alpha }\cos\left(\frac{\alpha \pi }{2}\right)+{\left(2{\omega \tau }_{\mathrm{CC}}\right)}^{2\alpha }}\right],$$
where there is a single time parameter τ_cc_, in contrast to distribution C(τ_c_) considered in the ME model, and a non-exponentiality of the dynamical process is introduced and controlled via the width parameter α. This model also described well the ^7^Li data, and therefore it was not possible for the authors to determine which model was more accurate based on their data.

Furthermore, Haaks et al. [184] investigated Li^+^ dynamics in 70Li_2_S-30P_2_S_5_ glass electrolyte (LiPS-GL) and glass-ceramic (LiPS-GC) obtained from this glass after heat treatment at 553 K for 1 h. They used ^7^Li T_1_ as well as static field gradient NMR to characterize short-range ion jumps and long-range ion diffusion, respectively. Although, in general, ceramization involves controlled crystallization and produces materials featuring both amorphous and crystalline phases, none of the ^7^Li NMR approaches yielded evidence for bimodal Li^+^ dynamics for LiPS-GC. Their results showed that ceramization substantially enhances the Li^+^ mobility on all length scales. They concluded that the correlation functions in LiPS-GC are strongly nonexponential and they used the same ME approach considered by Gabriel et al. [178] to describe the ^7^Li T_1_ profiles (Equation (34)).

The polymer electrolyte made of unentangled poly (propylene glycol) (PPG) and lithium perchlorate (LiClO_4_) or LiTFSI has been investigated by Becher et al. [185]. They used ^1^H and ^7^Li T_1_ measurements as a function of frequency at salts concentrations of 15:1 and 6:1 (ratio of polymer oxygens to lithium ions) to study the polymer and Li^+^ dynamics, respectively, in the wide range of temperatures of 290–380 K. The R_1_ profiles of the polymer were successfully described by assuming that segmental motion at short times ($R_{1\mathrm{CD}}\left(\omega \right)$) was followed by Rouse dynamics at longer times ($R_{1\mathrm{Rouse}}\left(\omega \right)$), omitting further intermolecular (translational) contributions. Thus, the total relaxation rate was depicted by a weighted superposition of these two mechanisms:(37)$$R_{1}^{\mathrm{Polymer}}\left(\omega \right)=R_{1\mathrm{CD}}\left(\omega \right)+R_{1\mathrm{Rouse}}\left(\omega \right).$$

The segmental motion was treated in the framework of glassy dynamics and the related spectral density showed the CD form (Equation (17)). On the other side, the contribution associated with Rouse dynamics was given by [188]:(38)$$R_{1\mathrm{Rouse}}\left(\omega \right)=A_{\mathrm{DD}}\tau_{O}\left[\overset{N-1}{\underset{p,q=1}{\sum }}\frac{p^{2}+q^{2}}{{\left(p^{2}+q^{2}\right)}^{2}+{\left({\omega \tau }_{o}N^{2}\right)}^{2}}+4\overset{N-1}{\underset{p,q=1}{\sum }}\frac{p^{2}+q^{2}}{{\left(p^{2}+q^{2}\right)}^{2}+{\left(2{\omega \tau }_{o}N^{2}\right)}^{2}}\right],$$
where N denotes the number of Rouse beads and τ_o_ is a time constant which is proportional to the friction coefficient of the Rouse model reduced. Concerning polymer segmental motion, the addition of salts leads to not only an increase of correlation times but also of dynamical heterogeneity. Even bimodal dynamical behavior was observed for the 15:1 salt concentration, which was explained by the coexistence of salt-rich and salt-depleted regions in these samples. For this concentration, the authors described the ^1^H relaxation dispersion by a superposition of two CD functions. On the other hand, they concluded that there was a strong coupling of polymer and Li^+^ dynamics and therefore described the ^7^Li relaxation profiles using the same model given by Equation (37). Interestingly, bimodal dynamics were not observed for Li^+^, which suggested that the vast majority of Li^+^ accumulate in salt-rich domains, whereas a very minor fraction reside in salt-depleted domains so that, unlike for the polymer segmental motion, different Li^+^ dynamics in the latter regions remain unobserved.

## 6. Conclusions

In the last few decades, researchers have focused on the development and improvement of electrolyte materials. Irrespective of the increasing complexities of materials’ design through stimuli such as temperature, materials composition and concentration, relaxometry measurements have revealed how the various ion interactions affect the electrolytes’ dynamics. From liquids to solids, relaxometry measurements have provided pivotal information that has advanced materials design and optimization. For example, studies of solute particles in ionic liquids have demonstrated the dissimilarities between the polar and non-polar local environments. In deep eutectic solvents, aluminum relaxometry revealed aluminum species-specific dynamics that were dependent on both the aluminum salt concentration and the hydrogen bond donor conformational flexibility. In confined liquid electrolytes, relaxometry detailed well-pronounced frequency-dependent dispersion profiles that are distinct for interactions between ions and the confining host material.

Relaxometry has also provided evidence of the fully solvated free ion nature of Mg^2+^ irrespective of the concentration in the MgTFSI/DME electrolyte. Moreover, it has revealed the existence of two different ion populations (“bulk-like” and “wall-interacting” fractions) in specific types of confined IL electrolytes. Despite the stronger interactions associated with solid electrolytes, relaxometry has provided evidence of the multiple ionic environments (i.e., mobile vs. immobile) that exist, as well as the dissimilarities between their cationic and anionic dynamics. Finally, relaxometry results have shown that ceramization substantially enhances Li^+^ mobility in glass–ceramic electrolytes. The results from all these studies are clear evidence of the potential of NMR relaxometry to study dynamics in electrolytes for energy storage applications.

## Data Availability

Data sharing not applicable. No new data were created or analyzed in this study. Data sharing is not applicable to this article.

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
