# Peer review of "Nuclear Magnetic Resonance Relaxation Pathways in Electrolytes for Energy Storage"

_ijms, 2023, doi:10.3390/ijms241210373_

Round 1
Reviewer 1 Report
The current work focuses on the Nuclear Magnetic Resonance Relaxation Pathways in Electrolytes 2 for Energy Storage. The experimental work appears to have been carried out well. However, a few points deserve attention for further publication. I suggest that it is accepted for publication after the following revisions:
- What is the importance of carrying out this study? What is the difference between this study and other studies published recently in the literature? What kind of problems does this technique present? What are the advantages? Issues explored? These questions should be explained in the manuscript abstract.
- The definition of Nuclear Magnetic Resonance Relaxation must be better defined in the manuscript.
- A topic exploring the Problems of Nuclear Magnetic Resonance Relaxation during preparation must be inserted in the manuscript;
- Problems in handling Nuclear Magnetic Resonance Relaxation need to be discussed in the manuscript;
- A scheme with the Nuclear Magnetic Resonance Relaxation technique must be inserted in the manuscript;
- A topic about Nuclear Magnetic Resonance Relaxation this discussion must be inserted in the manuscript;
- How does the presence of additives affect the preparation of Nuclear Magnetic Resonance Relaxation? These discussions must be inserted in the manuscript;
- A current trend is the Preparation of Nuclear Magnetic Resonance Relaxation. This discussion needs to be included in the manuscript.
- Nuclear Magnetic Resonance Relaxation - The manuscript needs to include aspects related to this strategy.
- What are the main applications of Nuclear Magnetic Resonance Relaxation? Authors should cite recent examples from the literature.
- What are the prospects for Nuclear Magnetic Resonance Relaxation to be applied on an industrial scale? What are the challenges to be overcome? These discussions need to be included in the manuscript.
- The authors could clarify the mechanism, advantages, problems, and solutions for the Nuclear Magnetic Resonance Relaxation systems in the manuscript's abstract.
- In addition, authors should highlight the advantages/disadvantages of these Nuclear Magnetic Resonance Relaxation methods for industrial application and how this information will be addressed in the manuscript.
- Advantages for Nuclear Magnetic Resonance Relaxation: Which methods have advantages? Are they simple ways to contribute? When compared with other sustainable techniques? Authors must leave this clear information to throughout the text and the procedures discussed in this manuscript. In addition, this information is needed for Nuclear Magnetic Resonance Relaxation systems contribution protocols to be applied on an industrial scale.
- Problems with Nuclear Magnetic Resonance Relaxation Systems: Does this protocol have a significant problem? This discussion could be improved.
- Additionally, advances in the Nuclear Magnetic Resonance Relaxation systems with engineered tailor-made have been performed with other strategies. It may open new opportunities. This discussion could be improved.
- This review broached interest in the progress and recent applications of Nuclear Magnetic Resonance Relaxation: The main contributions to the accomplishment of this work must be included in the conclusion.
- Please check all references according to the author's instructions.
- Include more details in the figures (error bars) and tables captions.
- The manuscript must be formatted according to the journal's standards.
Minor editing of the English language required
Reviewer 2 Report
I have reviewed a version of this manuscript previously for another journal, and had found it appropriate for publication previously. This work is a tiny review of NMR relaxation mechanisms in electrolytes, with relevance to energy science, and provides a much needed overview.
The review provides a synthesis of relevant NMR theory with a review of what particular literature already exists on NMR studies of ionic liquids and deep eutectic solvents. Therefore, the review will be of relevance to both NMR researchers and those who work in the IL and DES fields. Upon reading the manuscript again, a minor comment came up: The figure captions 1, 2, 4, 5, 6 should specify more NMR experimental parameters, in particular the magnetic field at which the studies were performed so that readers do not need to refer to the cited paper in order to understand what is shown in the figures.
